# A Fast, Hybrid, Time-Domain Discontinuous Galerkin-Physical Optics Method for Composite Electromagnetic Scattering Analysis

Ceyi Ma [1,2], Yinghong Wen [1,2] and Jinbao Zhang [1,2,*]

1 Institute of Electromagnetic Compatibility, Beijing Jiaotong University, Beijing 100044, China; 18111054@bjtu.edu.cn (C.M.); yhwen@bjtu.edu.cn (Y.W.)
2 Beijing Engineering Research Center of EMC and GNSS Technology for Rail Transportation, Beijing 100044, China
* Correspondence: jbzhang@bjtu.edu.cn

**Abstract:** To accelerate the solution of transient electromagnetic scattering from composite scatters, a novel hybrid discontinuous Galerkin time domain (DGTD) and time-domain physical optics (TDPO) method is proposed. The DGTD method is used to solve the accurate scattering field of the multi-scale objects region, and a hybrid explicit-implicit time integration method is also used to improve the efficiency of multi-scale problems in the time domain. Meanwhile, the TDPO method is used to accelerate the speed of surface current integration in an electrically large region. In addition, the DGTDPO method considers the mutual coupling between two regions, and effectively reduces the number of numerical calculations for the other space of the composite target, thereby significantly reducing the computer memory consumption. Numerical results certified the high efficiency and accuracy of the hybrid DGTDPO. According to the results, in comparison with the DGTD algorithm in the entire computational domain, the DGTDPO method can reduce computing time and memory by 90% and 70% respectively. Meanwhile, the normalized root mean square deviation (NRMSD) of the time-domain, high-frequency approximation method is over 0.2, and that of the DGTDPO method is only 0.0971. That is, compared with the approximation methods, the hybrid method improves the accuracy by more than 64%.

**Keywords:** discontinuous Galerkin time-domain (DGTD) method; time domain physical optics (TDPO); hybrid algorithm; composite electromagnetic scattering





## 1. Introduction

Nowadays, analysis of multi-scale composite transient electromagnetic scattering is an essential part of ocean detection [1], the reconstructing of the microwave imaging of objects [2], etc. A scatterer system with complex geometry, multi-scale and multi-object characteristics requires higher calculation accuracy. Moreover, a large-scale environment, such as a rough sea or land surface, will lead to a sharp increase in computing memory and time, and computers cannot keep up. These problems bring challenges to computational electromagnetism.

In order to take into account both accuracy and efficiency, the main method is to mix the full wave method with the high frequency approximation algorithm. In these methods, the full-wave methods, such as the method of moments (MoM), the finite-element time domain (FETD), finite volume time domain (FVTD) and finite-difference time domain (FDTD) calculate the regions with fine structures well. High-frequency algorithms are adopted for the calculation of electrically large areas, so as to improve the calculation speed and reduce the memory costs. Due to the good accuracy of MoM, a class of hybrid methods of MoM and high-frequency approximation algorithms are widely used [3–8]. However, the MoM has the problem of low frequency breakdown [9–11]. Simultaneously,

as a result of the dense matrix, the MoM is slightly inferior to other time-domain algorithms for solving broadband problems in the time domain. Hence, the hybrid method of FDTD and PO was proposed in [12–15]. However, due to the ladder error in the calculation of unstructured targets and surface fitting, the accuracy of electromagnetic scattering problem of complex targets had to be improved. Furthermore, the combination of FVTD and PO had also been tried. One paper [16] discussed the 3D radar cross section (RCS) of a sphere over a plate through FVTD/PO. This method achieves good accuracy for an electrically small sphere; and the larger the electrical size of the plate and the further the asymptotic and FVTD regions are, the higher accuracy of this method is. However, the hybrid method only calculates the frequency domain and does not involve the time domain and multi-scale problems. In addition, according to the conclusion in the literature [17], the DGTD method requires more time and memory than the FVTD algorithm, but it is able to handle multi-scale structures well; the FVTD method is not suitable for electrically large problems.

In the calculations of the transient scattering of multi-scale targets, a full-wave method named discontinuous Galerkin time domain (DGTD) [18–23] has more advantages than other methods in solving broadband multi-scale problems. With respect to geometric modeling, the multi-scale structure can be divided into several subdomains through DGTD. Furthermore, the grid density of each subdomain can be adjusted flexibly. As a result of the uneven tetrahedral grid technology, DGTD is much more accurate than FDTD. Moreover, DGTD is able to divide a large system's matrix into a group of smaller matrices, which means DGTD is more efficient than FETD at calculating complex structures. In terms of time integration, DGTD can utilize explicit and implicit hybrid schemes in various subdomains and accelerate the solving process [24]. The flexibility of spatial and temporal discretization makes the DGTD method efficient in multi-scale problems. However, when calculating the complex transient scattering of a complex target above a large area, such as rough sea surface or land through DGTD, the cost in terms of calculations is huge, and the requirement of memory cannot be borne by computers.

With the aim of quickly calculating the transient electromagnetic scattering of electrically large targets, time domain high-frequency techniques, such as time domain physical optics (TDPO) and time domain shooting and bouncing ray (TDSBR), have been developed. These techniques can reduce the burden of memory and improve the calculation speed, greatly. For the TDPO method, the only need is to calculate the induced electromagnetic current on the surface of the target, and then integrate it. Although it is difficult to deal with the multiple reflections [25–28], its accuracy is also acceptable if the surface is large enough and located in a far-field region. The TDSBR method combines the advantages of the geometric optics (GO) method and PO method [29–32], so the accuracy of TDSBR is better than that of TDPO. However, at least 10 ray tubes per wavelength are needed to build a TDSBR instrument. As the frequency increases, the computational complexity and time required for TDSBR increase. Although the time-domain, high-frequency approximation methods can speed up the solution, their accuracy for complex problems such as multi-scale structures is still poor.

According to the characteristics of the above methods, the DGTD method has more advantages than FDTD, FETD and FVTD in complex geometry modeling, time domain integration and accuracy, and TDPO is simpler and faster than TDSBR. Consequently, in this paper, a hybrid DGTD/TDPO method is proposed to improve the efficiency of the DGTD method in the calculation of transient electromagnetic scattering from multi-scale composite targets. The DGTDPO hybrid method avoids the step error of the FDTD/PO hybrid method when solving unstructured grid problems. At the same time, compared with the MoM/PO hybrid method, this hybrid method can avoid the low frequency breakdown of MoM. In fact, the DGTDPO hybrid method is not only valuable in the area of electromagnetic scattering, but also a great prospect for the study of electrochemical hydrogen evolution [33], the buckling and flexural vibration analysis of plates with intermediate supports [34] and the nonlinear vibration of fluid flow in single-walled carbon nanotubes [35].

In this work, a discontinuous Galerkin time domain solver (DGTD) is hybridized with the TDPO method for the first time to analyze transient multi-scale and composite objects' scattering problems. The proposed hybrid method possesses the advantages of both DGTD and TDPO. Numerical results demonstrate that the accuracy and efficiency of the proposed method are good enough in various scattering scenarios. The paper is organized as follows. The formulations of DGTDPO hybrid method are reported in Section 2. Numerical results aimed at validating the advantages of proposed hybrid method for multi-scale and multi-targets composite transient electromagnetic scattering are provided in Section 3. Finally, conclusions are drawn in Section 4.

## 2. Formulation of the Hybrid Method

As shown in Figure 1, the whole solution domain consists of two parts: the exact solution region (DGTD region) and the asymptotic domain (TDPO region). In the first region, a multi-scale and multi-object complex region is defined. As a result of the complex structure and requirements of higher precision in the first region, the DGTD method is adopted. TDPO is used to calculate the scattering field generated by asymptotic region itself and the transient excitation of DGTD region in the second region, which includes an electrically large plate or rough sea surface.

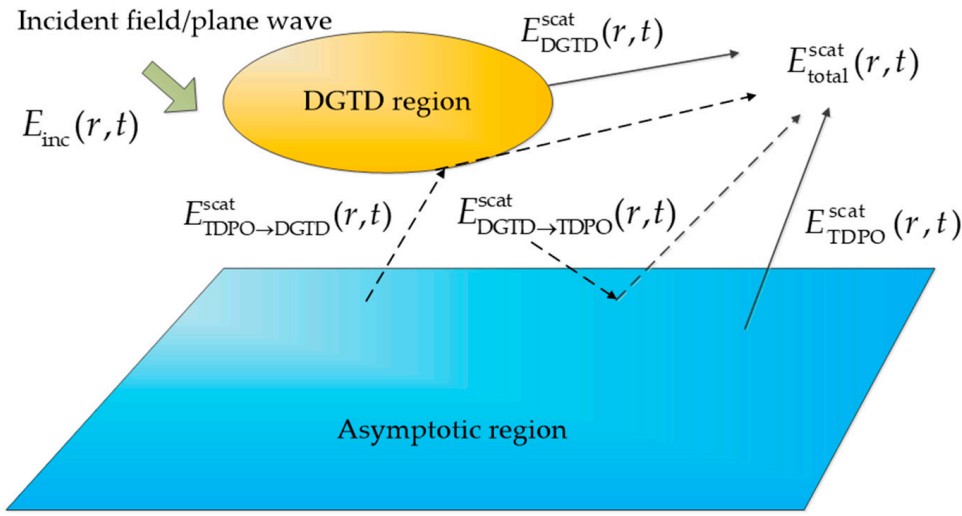

**Figure 1.** Combined contributions of the two regions.

The whole computational domain is illuminated through a transient source $\vec{E}_{\text{inc}}(r,t)$. The scattered field of whole region in far-field area is defined as $\vec{E}_{\text{total}}^{\text{scat}}(r,t)$, and approximately given by the following components:

$$\vec{E}_{\text{total}}^{\text{scat}}(r,t) \cong \vec{E}_{\text{DGTD}}^{\text{scat}}(r,t) + \vec{E}_{\text{TDPO}}^{\text{scat}}(r,t) + \vec{E}_{\text{DGTD}\rightarrow\text{TDPO}}^{\text{scat}}(r,t) + \vec{E}_{\text{TDPO}\rightarrow\text{DGTD}}^{\text{scat}}(r,t) \quad (1)$$

The first two terms $\vec{E}_{\text{DGTD}}^{\text{scat}}(r,t)$ and $\vec{E}_{\text{TDPO}}^{\text{scat}}(r,t)$ are the primary scattered fields generated from DGTD region and TDPO region, respectively. Nevertheless, both terms do not contain the multiple reflections between two regions. These interactions are included in secondary scattered fields denoted as $\vec{E}_{\text{DGTD}\rightarrow\text{TDPO}}^{\text{scat}}(r,t)$ and $\vec{E}_{\text{TDPO}\rightarrow\text{DGTD}}^{\text{scat}}(r,t)$. $\vec{E}_{\text{DGTD}\rightarrow\text{TDPO}}^{\text{scat}}(r,t)$ represents the scattered field through the asymptotic region when illuminated through the field generated by the DGTD region. Meanwhile, $\vec{E}_{\text{TDPO}\rightarrow\text{DGTD}}^{\text{scat}}(r,t)$ is defined as the inverse process of $\vec{E}_{\text{DGTD}\rightarrow\text{TDPO}}^{\text{scat}}(r,t)$. Additionally, we assume that the asymptotic region is significantly larger than the DGTD region. At the same time, because

with the increase of the distance between two regions, the calculation accuracy will be improved [16], the distance between the two regions has to be large enough.

### 2.1. Scattered Field Generated by the DGTD Region

The primary scattering contribution from the DGTD region is computed by using the DGTD method. Let $\Omega \subset R^3$ be the calculation area enclosing a scatterer, and $n$ the unitary outward normal to its boundary $\partial\Omega$. Define $\Omega_h$ as a discretization of $\Omega$; $\Omega_h$ consists of non-overlapping tetrahedral elements, and $\Omega_h = \cup_{m=1}^{N}\Omega_m$, where $N \in N^*$ is the number of mesh elements.

As shown in the Figure 2, the inner surfaces of the discretization are denoted by $\partial\Omega_{mk} = \partial\Omega_m \cap \partial\Omega_k$; $\Omega_k$ are adjacent cells of $\Omega_m$, and $\vec{n}_{mk}$ is defined as the unit vector normal to the surface $\partial\Omega_{mk}$ oriented from $\Omega_m$ toward $\Omega_k$.

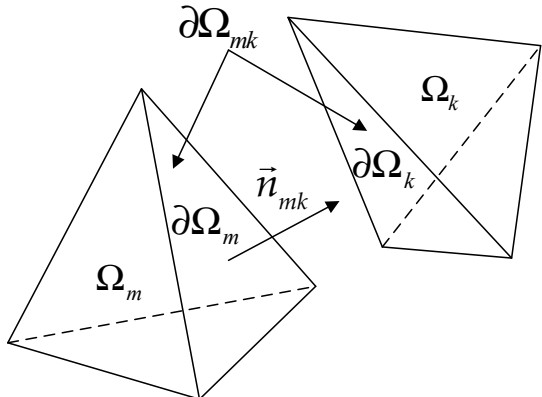

**Figure 2.** Spatially discrete elements in the discontinuous Galerkin time domain (DGTD) method.

Applying the discontinuous Galerkin testing to Maxwell curl equations in $\Omega_m$ yields [20]:

$$\iiint_{\Omega_m} \mu \frac{\partial H}{\partial t} \cdot N_{q'}^m dv + \iiint_{\Omega_m} \sigma_h H_m \cdot N_{q'}^m dv + \iiint_{\Omega_m} (\nabla \times E_m) \cdot N_{q'}^m dv$$
$$= \oiint_{\partial\Omega_{mk}} k_e^m [n \times (E_m - E_k) - M_s] \cdot N_{q'}^m ds + \oiint_{\partial\Omega_{mk}} v_h^m \{n \times [n \times (H_k - H_m) - J_s]\} \cdot N_{q'}^m ds \tag{2}$$

$$\iiint_{\Omega_m} \varepsilon \frac{\partial E}{\partial t} \cdot N_{q'}^m dv + \iiint_{\Omega_m} \sigma_e E_m \cdot N_{q'}^m dv + \iiint_{\Omega_m} (\nabla \times H_m) \cdot N_{q'}^m dv$$
$$= \oiint_{\partial\Omega_{mk}} k_h^m [n \times (H_k - H_m) - J_s] \cdot N_{q'}^m ds + \oiint_{\partial\Omega_{mk}} v_e^m \{n \times [n \times (E_m - E_k) - M_s]\} \cdot N_{q'}^m ds \tag{3}$$

where $\varepsilon$, $\mu$ and $\sigma$ are the permittivity, permeability and conductivity of the $\Omega$ respectively. In addition, $J_s$ and $M_s$ are surface electric current and magnetic current at the interface of adjacent elements, respectively. Moreover, $N_{q'}^m$ is the edge basis function of the $m$-th tetrahedral element. Meanwhile, the terms $k_e^m$, $k_h^m$, $v_e^m$ and $v_h^m$ are defined as the numerical flux factors in Table 1, and the subscript $k$ represents elements adjacent to $m$ elements.

**Table 1.** Three forms of numerical flux in the DGTD method.

| Numerical Flux Factors | $k_e^m$ | $k_h^m$ | $v_h^m$ | $v_e^m$ |
|---|---|---|---|---|
| Centered numerical flux | 1/2 | 1/2 | 0 | 0 |
| Upwind numerical flux | $\frac{Y^{mk}}{Y^m + Y^{mk}}$ | $\frac{Z^{mk}}{Z^m + Z^{mk}}$ | $\frac{1}{Y^m + Y^{mk}}$ | $\frac{1}{Z^m + Z^{mk}}$ |
| Partially penalized numerical flux | $\frac{Y^{mk}}{Y^m + Y^{mk}}$ | $\frac{Z^{mk}}{Z^m + Z^{mk}}$ | $\frac{\tau}{Y^m + Y^{mk}}$ | $\frac{\tau}{Z^m + Z^{mk}}$ |

Numerical flux is led into at the common interface of adjacent elements—that is, the tangential component of electromagnetic field on the shared surface of two adjacent elements is no longer continuous. We assume that a multi-scale structure is divided into subdomains; the semi-discretized system of equations by the DGTD method will be [20]:

$$\mu\{M\}\frac{dH_m}{dt} + \sigma_h\{M\}H_m + \{S\}E_m + \left\{F_{ke}^k\right\}E_m - \{F_{ke}\}E_m + \left\{G_{vh}^k\right\}H_k - \{G_{vh}\}H_m + M_{sk} - J_{sv} = 0 \tag{4}$$

$$\varepsilon\{M\}\frac{dE_m}{dt} + \sigma_e\{M\}E_m - \{S\}H_m - \left\{F_{kh}^k\right\}H_k + \{F_{kh}\}H_m + \left\{G_{ve}^k\right\}E_k - \{G_{ve}\}E_m + M_{sv} + J_{sk} = 0 \tag{5}$$

$$\begin{cases} J_{sk,q\prime} = k_h^m \oiint_{\partial\tau_m} J_s \cdot N_{q\prime}^m ds, & J_{sv,q\prime} = v_h^m \oiint_{\partial\tau_m} (n \times J_s) \cdot N_{q\prime}^m ds \\ M_{sk,q\prime} = k_e^m \oiint_{\partial\tau_m} M_s \cdot N_{q\prime}^m ds, & M_{sv,q\prime} = v_e^m \oiint_{\partial\tau_m} (n \times M_s) \cdot N_{q\prime}^m ds \end{cases} \tag{6}$$

where $\{M\}$ is the mass matrix; $\{S\}$ is the stiffness matrix; $\left\{F_{kh}^k\right\}$, $\{F_{kh}\}$, $\left\{G_{ve}^k\right\}$ and $\{G_{ve}\}$ are the flux matrices. $E_m$ and $H_m$, and $E_k$ and $H_k$ are the edge electric field and magnetic field of the $m$-th element. The superscript $k$ represents adjacent elements of $m$ elements. The above matrix $\{M\}$ and $\{S\}$ elements are:

$$M_{q\prime,q} = \iiint_{\tau_m} N_{q\prime}^m \cdot N_q^m dv, \qquad S_{q\prime,q} = \iiint_{\tau_m} N_{q\prime}^m \cdot \left(\nabla \times N_q^m\right) dv, \qquad q, q\prime = 1, 2, \dots, 6 \tag{7}$$

and flux matrices $\left\{F_{kh}^k\right\}$, $\{F_{kh}\}$, $\left\{G_{ve}^k\right\}$ and $\{G_{ve}\}$ are:

$$\begin{cases} F_{kh,q\prime q} = k_h^m \oiint_{\partial\tau_m} N_{q\prime}^m \cdot \left(n \times N_q^m\right) ds, & F_{ke,q\prime q} = k_e^m \oiint_{\partial\tau_m} N_{q\prime}^m \cdot \left(n \times N_q^m\right) ds \\ G_{vh,q\prime q} = v_h^m \oiint_{\partial\tau_m} N_{q\prime}^m \cdot \left(n \times n \times N_q^m\right) ds, & G_{ve,q\prime q} = v_e^m \oiint_{\partial\tau_m} N_{q\prime}^m \cdot \left(n \times n \times N_q^m\right) ds \\ F_{kh,q\prime q}^k = k_h^m \oiint_{\partial\tau_m} N_{q\prime}^m \cdot \left(n \times N_p^{m+}\right) ds, & F_{ke,q\prime q}^k = k_e^m \oiint_{\partial\tau_m} N_{q\prime}^m \cdot \left(n \times N_p^{m+}\right) ds \\ G_{vh,q\prime q}^k = v_h^m \oiint_{\partial\tau_m} N_{q\prime}^m \cdot \left(n \times n \times N_p^{m+}\right) ds, & G_{ve,q\prime q}^k = v_e^m \oiint_{\partial\tau_m} N_{q\prime}^m \cdot \left(n \times n \times N_p^{m+}\right) ds \end{cases} \tag{8}$$

where $q, q\prime, p = 1, 2, \dots, 6$, and $N_{q\prime}^{m+}$ is the m-th adjacent element basis function.

In this paper, the hybrid explicit-implicit time integration method [7] is used to calculate the multi-scale problems. In this method, the implicit time integration scheme is locally applied in the fine region of the grid, while the explicit time scheme is retained in the complementary part, which can further improve the stability of the calculation and reduce the amount of calculation. As shown in Figure 3, the perfect match layer (PML) is used as boundary condition. We utilize a time-domain near-field to far-field (NFFF) transformation method from [36], and calculate the scattered electric and magnetic fields $\overset{\rightarrow\text{scat}}{E}_{\text{DGTD}}(r,t)$ and $\overset{\rightarrow\text{scat}}{H}_{\text{DGTD}}(r,t)$ in far field. In order to compute the NFFF transformation, a proper Huygens surface is considered to be in the DGTD region, and then, the time-domain scattering field in DGTD region can be obtained.

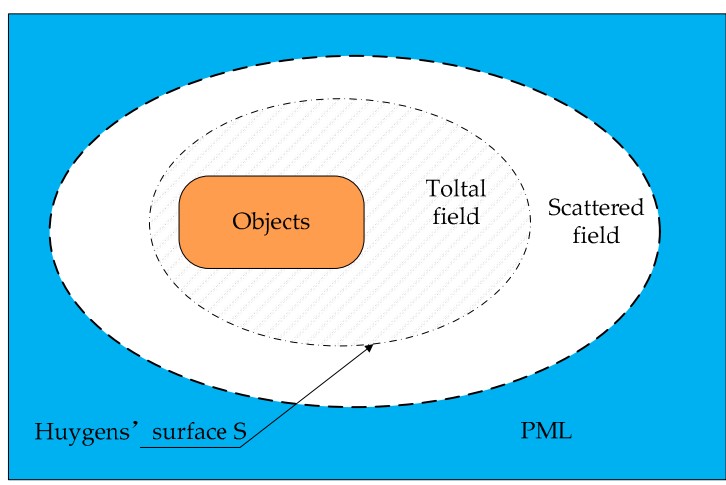

**Figure 3.** DGTD region.

*2.2. Scattered Field Generated by the TDPO Region*

The primary scattered field generated by the asymptotic region is calculated through the TDPO method. The TDPO method assumes that the scattered field is only generated from the tangent plane of each point on the geometric illumination side of the object, while the field is zero in the shadow area of the object [37]. Therefore, the TDPO method can be used to solve the scattering field of a large-scale perfect electrically conducting (PEC) object without considering the creeping wave.

We can obtain the approximate surface current density distribution $\vec{J}_{rs}(R, t)$ excited by the transient source:

$$\vec{J}_{rs}^{\text{TDPO}}\left(\vec{R}, t\right) = \begin{cases} 2\hat{n} \times \vec{H}_{\text{inc}}\left(\vec{R}, t\right) \\ 0 \end{cases} \tag{9}$$

According to the surface current density of the solution area defined by Equation (9), the computation of the scattered field is simplified as an integral over the Green's function:

$$\vec{H}_{\text{TDPO}}^{\text{scat}}\left(\vec{R}, t\right) \approx -\frac{1}{4\pi Rc}\hat{a}_R \times \iint_{s\prime} \frac{\partial \vec{J}_{rs}^{\text{TDPO}}\left(\vec{R}\prime, t\right)}{\partial t}dS\prime \tag{10}$$

$$\vec{E}_{\text{TDPO}}^{\text{scat}}\left(\vec{R}, t\right) \approx -\frac{\eta_0}{4\pi Rc} \iint_{s\prime} \frac{\partial \vec{J}_{rst}^{\text{TDPO}}\left(\vec{R}\prime, t\right)}{\partial t}dS\prime \tag{11}$$

where

$$\vec{J}_{rst}^{\text{TDPO}}\left(\vec{R}\prime, t\right) = \vec{J}_{rs}^{\text{TDPO}}\left(\vec{R}\prime, t\right) - \left(\vec{J}_{rs}^{\text{TDPO}}\left(\vec{R}\prime, t\right) \cdot \hat{a}_R\right)\hat{a}_R \tag{12}$$

The scattered electric field $\vec{E}_{\text{DGTD}}^{\text{scat}}(r, t)$ can be solved through (5), and $\vec{E}_{\text{TDPO}}^{\text{scat}}(r, t)$ is solved by (9) and (11).

$$\vec{E}_{\text{TDPO}}^{\text{scat}}\left(\vec{R}, t\right) \approx -\frac{\eta_0}{4\pi Rc} \cdot$$
$$\iint_{S\prime} \frac{\partial 2\hat{n} \times \vec{H}_{\text{inc}}\left(\vec{R}\prime, t - \left(\left|\vec{R} - \vec{R}\prime\right| - \left|\vec{R}\prime\right|\right)/c\right) - \left(2\hat{n} \times \vec{H}_{\text{inc}}\left(\vec{R}\prime, t - \left(\left|\vec{R} - \vec{R}\prime\right| - \left|\vec{R}\prime\right|\right)/c\right) \cdot \hat{a}_R\right)\hat{a}_R}{\partial t}dS\prime \tag{13}$$

where $\vec{R}$ is the distance between the excitation source point and the integration points on the TDPO region's surface; $\vec{R}\prime$ is the distance between observation points and the integration points on TDPO region surface. The terms $\eta_0$ and $c$ are the wave impedance and light speed in free space, respectively. Assume that the TDPO region's surface $S\prime$ is discretized into $p$ triangular cells. Equation (14) represents the integral in (13) numerically:

$$\vec{E}_{\text{TDPO}}^{\text{scat}}\left(\vec{R}, t\right) \approx -\frac{\eta_0}{4\pi Rc} \cdot$$
$$\sum_p \left\{ 2\left[\hat{n} \times \frac{\partial\left(\vec{H}_{\text{inc}}\left(\vec{R}\prime, t - \left(\left|\vec{R} - \vec{R}\prime\right| - \left|\vec{R}\prime\right|\right)/c\right) - \left(\vec{H}_{\text{inc}}\left(\vec{R}\prime, t - \left(\left|\vec{R} - \vec{R}\prime\right| - \left|\vec{R}\prime\right|\right)/c\right) \cdot \hat{a}_R\right)\hat{a}_R\right)}{\partial t}\right]\Delta S_p \right\} \tag{14}$$

*2.3. Scattered Fields Generated by the Mutual Coupling between Two Regions*

In this paper, we suppose that the asymptotic region is located at the far field area of the DGTD region. The secondary scattered field includes two components. For the first term $\vec{E}_{\text{DGTD}\to\text{TDPO}}^{\text{scat}}\left(\vec{R}, t\right)$, we defined $\vec{H}_{\text{DGTD}}^{\text{scat}}(r, t)$ as an incident field of the TDPO

region. We combine it with Equations (5), (9) and (11) and obtain the coupling term $\overset{\rightarrow \text{scat}}{E}_{\text{DGTD} \to \text{TDPO}}\left(\overset{\rightarrow}{R}, t\right)$:

$$\overset{\rightarrow \text{scat}}{E}_{\text{DGTD} \to \text{TDPO}}\left(\overset{\rightarrow}{R}, t\right) \approx -\frac{\eta_0}{4\pi Rc} \iint_{S\prime} \frac{\partial 2\hat{n} \times \overset{\rightarrow \text{scat}}{H}_{\text{DGTD}}\left(\overset{\rightarrow}{R}\prime, t - \left(\left|\overset{\rightarrow}{R}_{\text{GP}}\right| - \left|\overset{\rightarrow}{R}_{\text{P}}\right|\right)/c\right) - \left(2\hat{n} \times \overset{\rightarrow \text{scat}}{H}_{\text{DGTD}}\left(\overset{\rightarrow}{R}\prime, t - \left(\left|\overset{\rightarrow}{R}_{\text{GP}}\right| - \left|\overset{\rightarrow}{R}_{\text{P}}\right|\right)/c\right) \cdot \hat{a}_R\right)\hat{a}_R}{\partial t} dS\prime \quad (15)$$

where $\overset{\rightarrow}{R}_{\text{GP}}$ is the distance between elements in the DGTD region and the TDPO region, as well as $\overset{\rightarrow}{R}_{\text{P}} = \left|\overset{\rightarrow}{R} - \overset{\rightarrow}{R}\prime\right|$. Then, (16) represents the integral in (15) numerically.

$$\overset{\rightarrow \text{scat}}{E}_{\text{DGTD} \to \text{TDPO}}\left(\overset{\rightarrow}{R}, t\right) \approx -\frac{\eta_0}{4\pi Rc} \cdot$$
$$\sum_p \left\{ 2\left[ \hat{n} \times \frac{\partial\left(\overset{\rightarrow \text{scat}}{H}_{\text{DGTD}}\left(\overset{\rightarrow}{R}\prime, t - \left(\left|\overset{\rightarrow}{R}_{\text{GP}}\right| - \left|\overset{\rightarrow}{R}_{\text{P}}\right|\right)/c\right) - \left(\overset{\rightarrow \text{scat}}{H}_{\text{DGTD}}\left(\overset{\rightarrow}{R}\prime, t - \left(\left|\overset{\rightarrow}{R}_{\text{GP}}\right| - \left|\overset{\rightarrow}{R}_{\text{P}}\right|\right)/c\right) \cdot \hat{a}_R\right)\hat{a}_R\right)}{\partial t}\right] \Delta S_p \right\} \quad (16)$$

Similarly, $\overset{\rightarrow \text{scat}}{E}_{\text{TDPO} \to \text{DGTD}}\left(\overset{\rightarrow}{R}, t\right)$ is an inverse process of $\overset{\rightarrow \text{scat}}{E}_{\text{DGTD} \to \text{TDPO}}\left(\overset{\rightarrow}{R}, t\right)$.

$$\overset{\rightarrow \text{scat}}{E}_{\text{TDPO} \to \text{DGTD}}\left(\overset{\rightarrow}{R}, t\right) \approx -\frac{\eta_0}{4\pi Rc} \cdot$$
$$\sum_G \left\{ 2\left[ \hat{n} \times \frac{\partial\left(\overset{\rightarrow \text{scat}}{H}_{\text{TDPO}}\left(\overset{\rightarrow}{R}\prime, t - \left(\left|\overset{\rightarrow}{R}_{\text{GP}}\right| - \left|\overset{\rightarrow}{R}_{\text{P}}\right|\right)/c\right) - \left(\overset{\rightarrow \text{scat}}{H}_{\text{TDPO}}\left(\overset{\rightarrow}{R}\prime, t - \left(\left|\overset{\rightarrow}{R}_{\text{GP}}\right| - \left|\overset{\rightarrow}{R}_{\text{P}}\right|\right)/c\right) \cdot \hat{a}_R\right)\hat{a}_R\right)}{\partial t}\right] \Delta S_G \right\} \quad (17)$$

where $\overset{\rightarrow}{R}_{\text{PG}}$ is the distance of elements between the TDPO region and DGTD region, and $S_G$ is the NFFF boundary.

According to the above equations, the DGTD region and the asymptotic region can be calculated separately, and then the scattering field generated by the coupling between the two regions can be calculated by using Equations (16) and (17), and finally the total scattering field of the whole solution region can be obtained through Equation (1).

## 3. Numerical Results

In this section, the DGTDPO method is used to calculate the RCS and scattering fields of three kinds of composite PEC scatterer, which include a sphere-plate model, a multi-scale object-plate model and a multi-scale, multi-target-rough sea surface model. The calculation time, memory consumption and accuracy of the DGTDPO method are analyzed and compared. In all composite scattering scenarios, the incident wave is a plane wave excited by a Gaussian pulse modulation:

$$\overset{\rightarrow}{E}_{\text{inc}}(r, t) = \hat{p} E_0 \mathbf{G}(t - k \cdot r/c_0) \quad (18)$$

$$\mathbf{G}(t) = \exp(-(t - t_0)^2/\tau_m^2) \cos(2\pi f_m(t - t_0)) \quad (19)$$

where $\hat{p}$ is the polarization; $k$ is the direction of propagation; $c_0$ is the speed of light. $E_0 = 1V/m$ is the amplitude; and $\mathbf{G}(t)$ is a Gaussian pulse with modulation frequency $f_m$ delay $t_0$, and duration $\tau_m$.

### 3.1. Sphere and Plate

In this example, as shown in the Figure 4, the scatterer is a composite object including a PEC sphere above a PEC plate residing in free space.

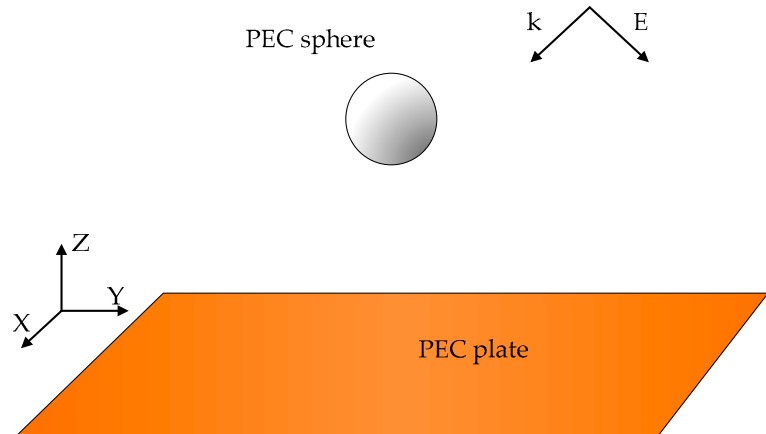

**Figure 4.** PEC sphere-plate composite object.

The PEC sphere radius $r_s$ is 1 m; the PEC plate's sides are $l_{px} = l_{py} = 10$m; and the distance $D$ between the sphere's center and the PEC plate's surface is 5 m. For the first simulation, the incident wave was a plane wave excited by a Gaussian pulse modulation. During the simulation, the transient scattered field of the composite object was computed by DGTDPO. All the MoM results were obtained by ANSYS HFSS-IE solver. In addition, HFSS and Mie are defined as the references; the results computed by various methods of normalized root mean square deviation (NRMSD) were used to estimate the deviation. NRMSD is defined as follows:

$$\text{NRMSD} = \frac{\sqrt{\frac{\sum_{m=1}^{n} (C(m) - R(m))^2}{n}}}{R(\max) - R(\min)} \tag{20}$$

where $m$ represents the value of the discrete result and $C(m)$ is one of results of other methods, and $R(m)$ is the result of HFSS.

As shown in Figure 5a, the scattering field of a PEC sphere computed by DGTD agrees well with Mie which is an analytic method and the NRMSD is equal to 0.0398. Figure 5b is computed by the Fourier-transformed and the NRMSD is 0.0288. Figure 5b illustrates that DGTD can calculate well for various electrical sizes. As shown in Figure 6a,b, these are the bistatic RCS results of the PEC sphere-plate composite object at 0.3004 GHz and 0.5 GHz, as well as the NRMSD is 0.0685 and 0.0632 respectively. The results are similar to these calculated by MoM. Although there are some deviations between 150° and 230°, the part below the PEC plate is not considered in the electromagnetic scattering calculation. Therefore, the accuracy of the hybrid algorithm can be guaranteed.

As shown in Table 2, because the DGTDPO hybrid method uses the TDPO method for large-scale background and the shadow region is automatically disregarded, the calculation speed and memory consumption are improved by 99.55% and 70.27% respectively, compared with DGTD.

**Table 2.** The CPU time and memory of DGTD and DGTDPO for a PEC sphere-plate composite object.

|  | DGTD | DGTDPO | Decreasing Rate |
| --- | --- | --- | --- |
| CPU Time | 60.87 h | 0.33 h | 99.46% |
| Memory | 1.79 G | 545 M | 70.27% |

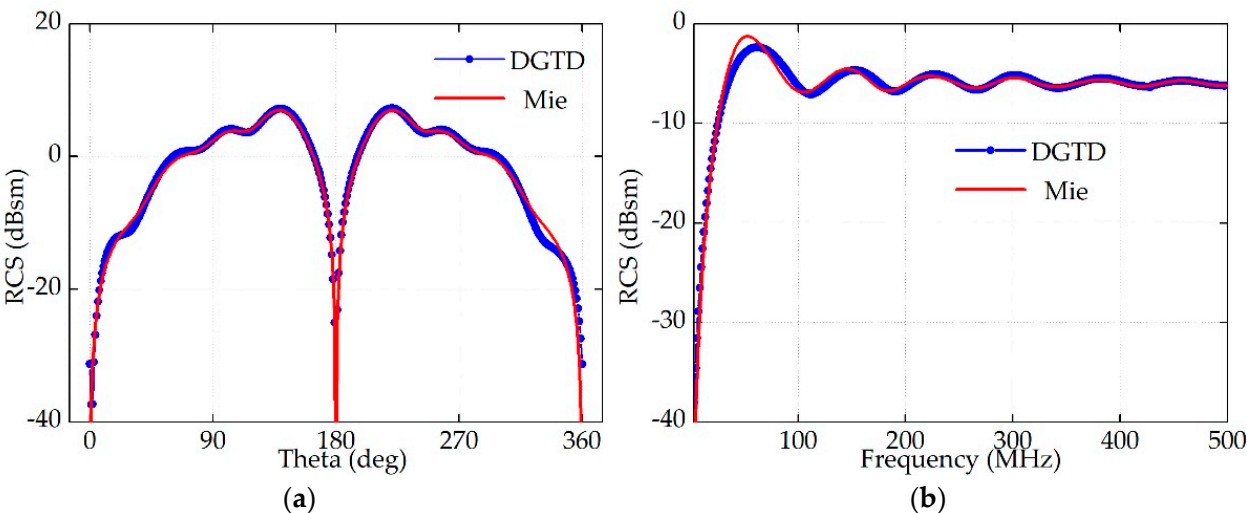

**Figure 5.** Bistatic radar cross section (RCS) of a perfect electrically conducting (PEC) sphere computed through DGTD and Mie at (**a**) 0.3004 GHz, and (**b**) the electric field of a PEC sphere computed through DGTD and Mie at range of frequencies from 0.1 to 500 MHz.

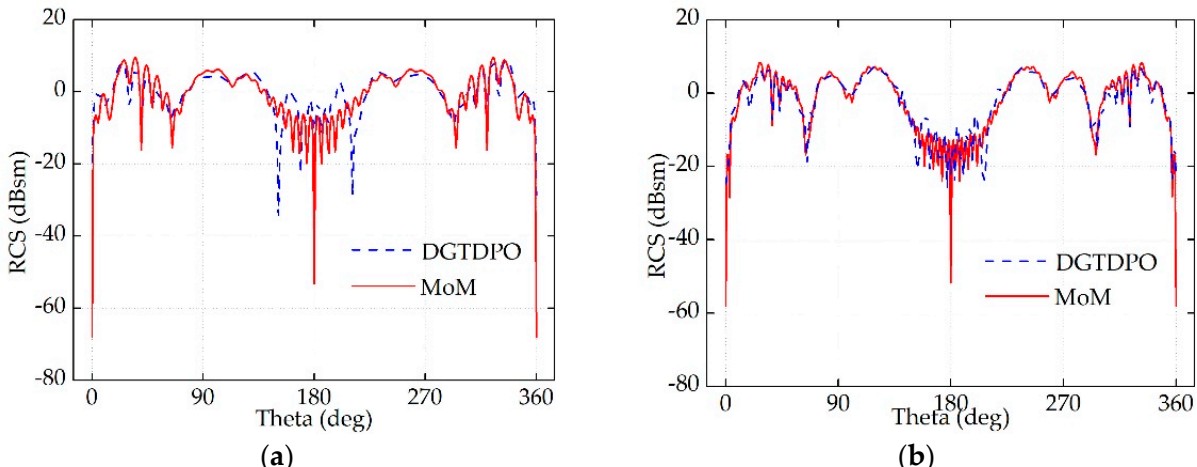

**Figure 6.** The RCS of a PEC sphere-plate computed at (**a**) 0.3004 GHz and (**b**) 0.5 GHz from the solutions of the DGTDPO method and Method of Moment (MoM).

### 3.2. Multi-Scale Object and Plate

The second scatterer was a multi-scale object above a PEC plate in Figure 7. The overall length of the object was 3.3 m, the tail size was 0.3 m and the thickness was 0.01 m. For the frequency range of the incident wave from 0.5 to 1 GHz, the object body was electrically large, but the tail was equivalent to an electrically small structure. Consequently, the scatterer can be considered as a multi-scale electromagnetic scattering problem.

As shown in the Figure 8a,b, the bistatic RCS results of an object at 0.5 GHz and 1 GHz through DGTD agree well with MoM, and the NRMSD values were 0.0427 and 0.0452 respectively. Meanwhile, in Figure 9a,b, the electric field in the time domain and frequency domain from 0.5 to 1 GHz was calculated by DGTD, and the result is almost consistent with that of HFSS; the resulting NRMSD values were 0.0112 and 0.0053 respectively. These results show that DGTD method can be used to calculate the multi-scale problem very well. At the same time, the accuracy of the above results is also an important guarantee for the hybrid algorithm's accuracy.

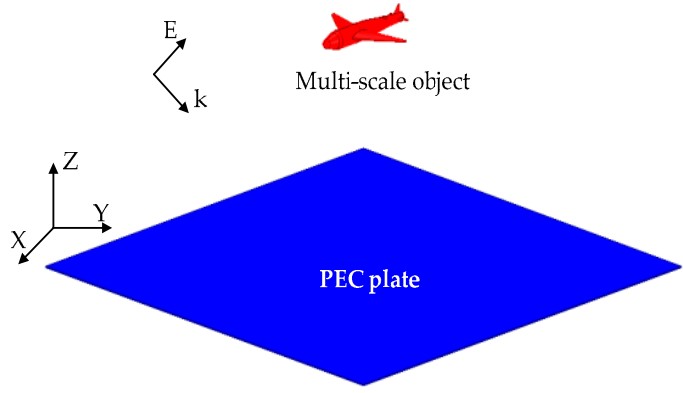

**Figure 7.** Multi-scale object-plate composite object.

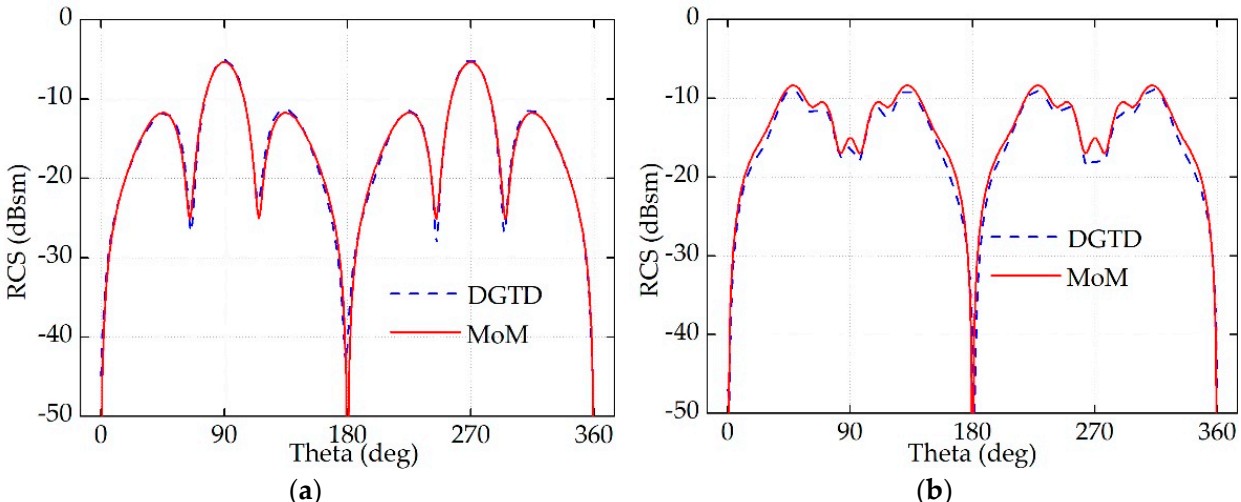

**Figure 8.** The RCS of a multi-scale object above a PEC plate computed at (**a**) 0.5 GHz and (**b**) 1 GHz from the solutions of the DGTD method and MoM.

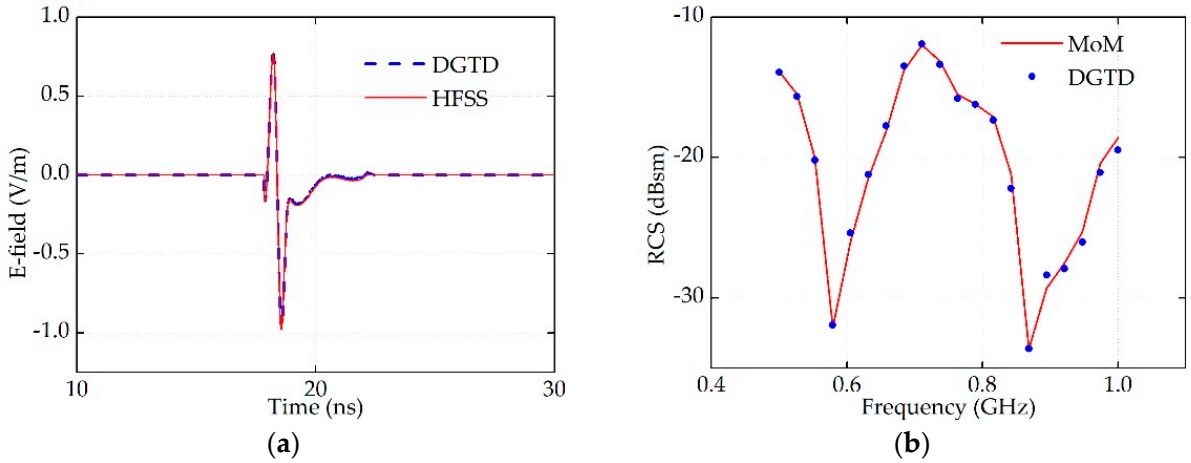

**Figure 9.** (**a**) The transient scattered electric field of a multi-scale object computed at theta = $0°$, phi = $0°$. (**b**) The RCS of a PEC object computed from 0.5 to 1GHz (theta = $0°$, phi = $0°$) from the solutions of the DGTD method and MoM.

Figure 10 indicates the superiority of the DGTD method in multiples scales over the time domain, high-frequency approximation methods, including TDPO, and the timedo-

main shooting and bouncing ray (TDSBR). The NRMSD of DGTD, TDPO and TDSBR are shown in Table 3.

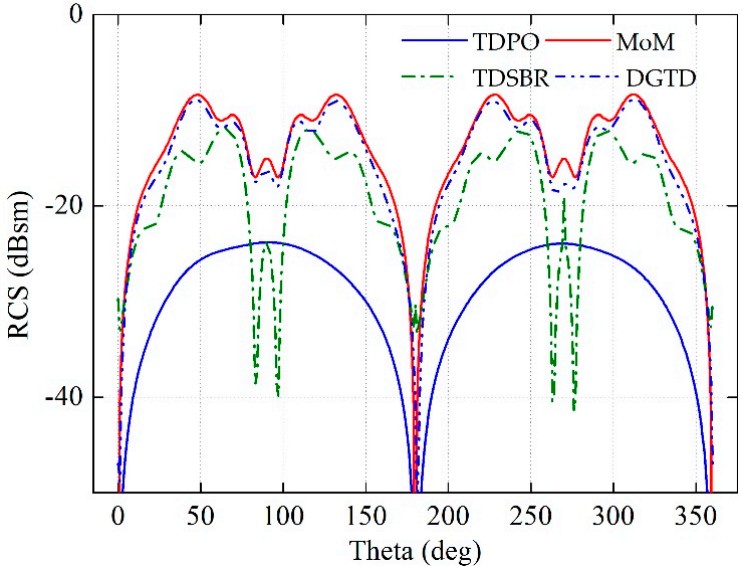

**Figure 10.** Accuracy comparison of DGTD, TDPO and TDSBR methods for RCS calculation at 1 GHz of multi-scale structural objects.

**Table 3.** The NRMSD of DGTD, TDPO and TDSBR for the multi-scale object at 1.5GHz.

| Method | DGTD | TDPO | TDSBR |
|--------|------|------|-------|
| NRMSD  | 0.0348 | 0.2360 | 0.1028 |

Compared with time domain high-frequency technique, Table 3 shows that DGTD has higher accuracy when solving multi-scale complex problems.

As shown in Figure 11a,b, these are the bistatic RCS results of the PEC sphere plate composite object at 1 and 1.5 GHz. In addition to the scattering field in the angle range below the rough sea surface, the DGTDPO results are similar to MoM.

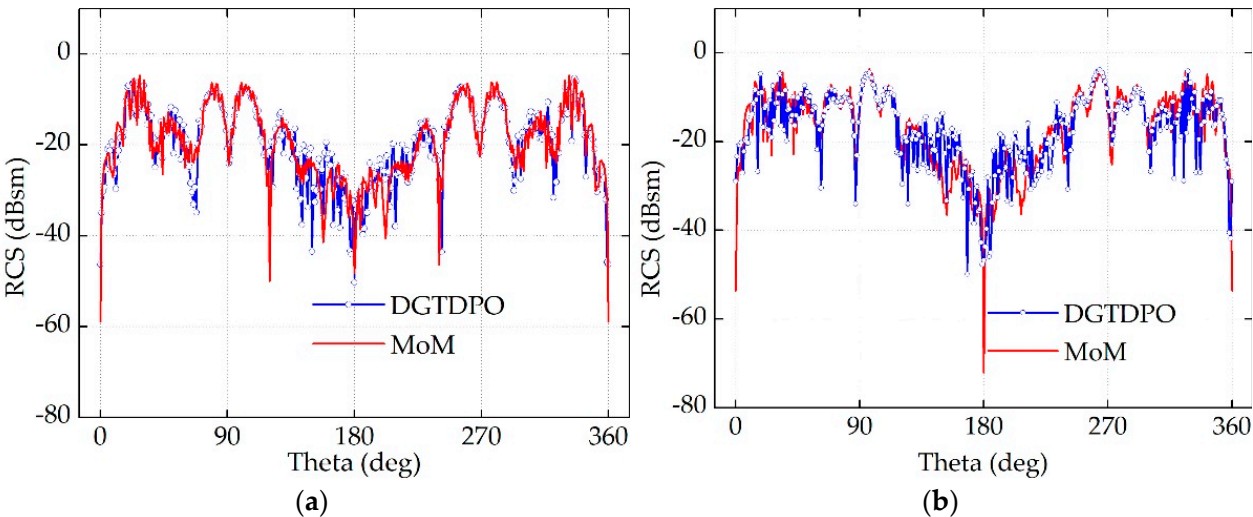

**Figure 11.** The RCS of a multi-scale object above a plate computed at (**a**) 1 GHz and (**b**) 1.5 GHz from the solutions of the DGTDPO method and MoM.

Compared with time domain high-frequency technique, Table 4 shows that DGTDPO has higher accuracy when solving for a multi-scale object than for plate composite objects.

**Table 4.** The NRMSD of DGTDPO, TDPO and TDSBR for multi scale-plate composite objects at 1.5 GHz.

| Method | DGTDPO | TDPO | TDSBR |
|---|---|---|---|
| NRMSD (1.5GHz) | 0.0823 | 0.1530 | 0.1431 |

As shown in Table 5, the complex multi-scale structure leads CPU time and memory both to get higher than the sphere-plate composited scatter. In addition, the calculation time and memory consumption of DGTDPO method were reduced by 99.54% and 72.56% respectively.

**Table 5.** The CPU time and memory of DGTD and DGTDPO for multi scale-plate composite objects.

| | DGTD | DGTDPO | Decreasing Rate |
|---|---|---|---|
| CPU Time | 315.56 h | 1.42 h | 99.54% |
| Memory | 5.46 G | 1.5 G | 72.56% |

### 3.3. Multi-Object and Rough Sea Surface

In the last example, a complex electromagnetic scattering scene of multiple objects above a rough sea surface is calculated and shown in Figure 12. Obviously, the electromagnetic scattering problem of multiple objects is more complex for than a single object. Furthermore, as a result of multiple coupling between the three multi-scale models as well as a rough surface, these will increase the computational complexity and difficulty of the hybrid algorithm.

$$S(k) = \frac{\alpha}{4|k|^3} \exp\left( -\frac{\beta g_c^2}{k^2 U_{19.5}^4} \right) \tag{21}$$

where $\alpha$ and $\beta$ are the dimensionless empirical constants, $\alpha = 8.10 \times 10^{-3}$ and $\beta = 0.74$; $g_c$ is the acceleration of gravity; $g_c = 9.81 \mathrm{m/s}^2$; and $U_{19.5}$ is the wind speed at 19.5 m above sea level [38].

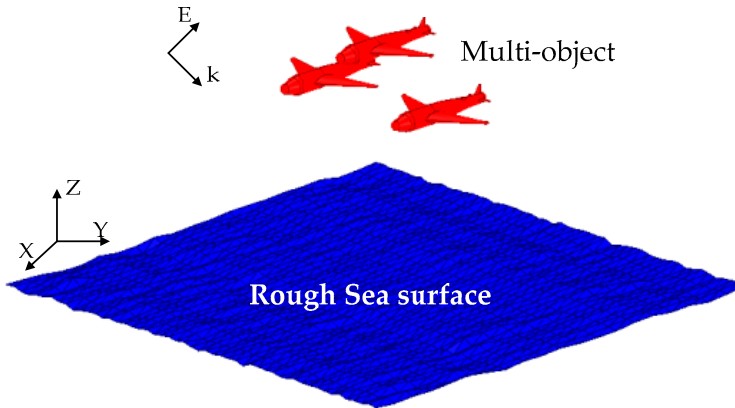

**Figure 12.** The multi-object-rough sea surface composite object.

As shown in the Figure 13a,b, the RCS of multiple objects at 0.5 GHz and 1 GHz were calculated via the DGTD method, and the NRMSD values were 0.0328 and 0.0619 respectively. These results demonstrate that DGTD method can also be used to calculate the multi-object problem.

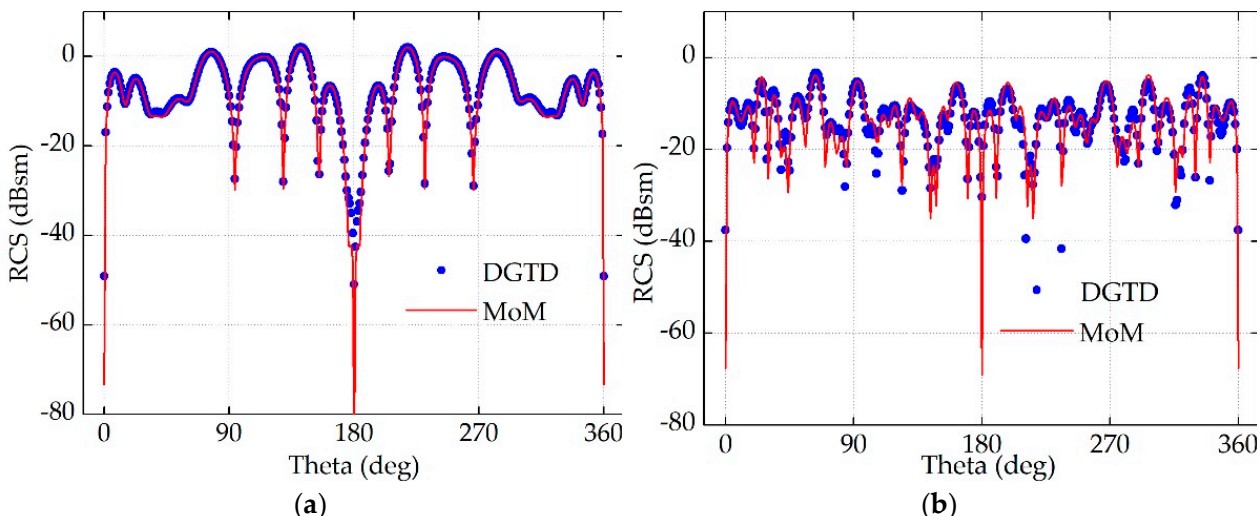

**Figure 13.** The RCS of multiple objects computed at (**a**) 1 GHz and (**b**) 1.5 GHz from the solutions of the DGTD method and MoM.

As shown in the Figure 14, the transient scattered electric field at theta = 0°, phi = 0° was calculated, and the NRMSD was 0.0312. This illustrates that the DGTDPO method calculates results for a multi-object-rough sea surface composite well in the time domain. Besides, Figure 15 shows the RCS of the multi-object and rough sea surface composite object at 1 GHz and 1.5 GHz calculated by DGTD method. These results are almost consistent with those of MoM. This shows that the DGTD method can also be used to calculate the electromagnetic scattering problem of the multi-object and rough sea surface composite.

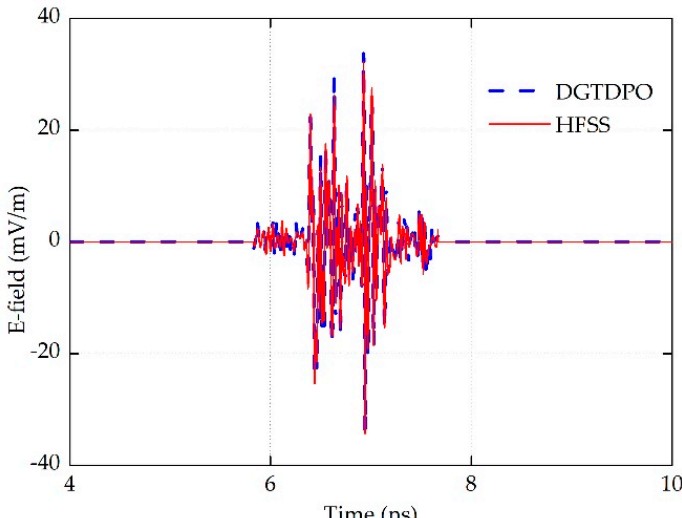

**Figure 14.** Transient scattered electric field of the multi-object-rough sea surface composite computed at theta = 0°, phi = 0°.

Table 6 shows that DGTDPO was more accurate than TDSBR and TDPO for the multi-object-rough sea surface composite.

**Table 6.** The NRMSD of DGTD, TDPO and TDSBR at 1.5GHz for multi-object-rough sea surface composite objects.

| Method | DGTDPO | TDPO | TDSBR |
|--------|--------|------|-------|
| NRMSD | 0.0971 | 0.2744 | 0.2003 |

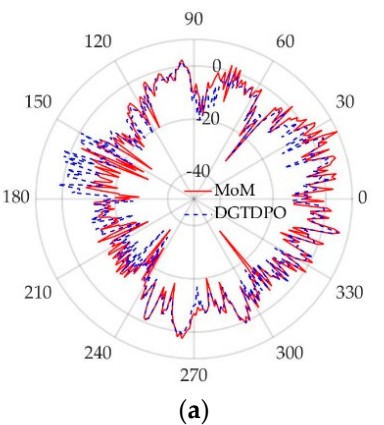 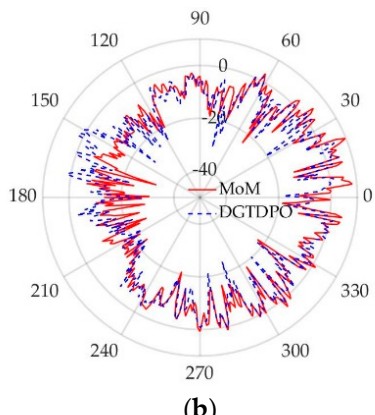

(**a**)                                            (**b**)

**Figure 15.** The RCS of the multi-object and sea surface composite computed at (**a**) 1 GHz and (**b**) 1.5 GHz from the solutions of the DGTDPO method and MoM.

As shown in Table 7, the complex multi-scale structure and rough sea surface bring about higher CPU time and memory requirements. Nevertheless, compared with DGTD, the calculation time and memory consumption of the DGTDPO method were decreased by 99.65% and 75.34% respectively.

**Table 7.** The CPU time and memory of DGTD and DGTDPO for multi-object-rough sea surface composite objects.

|  | DGTD | DGTDPO | Decreasing Rate |
|---|---|---|---|
| CPU Time | 1142.85 h | 4 h | 99.65% |
| Memory | 16.71 G | 4.12 G | 75.34% |

## 4. Conclusions

A hybrid DGTDPO method is proposed for accelerating the calculation of broadband scattering multi-scale and multi-object composite scatters. The hybrid method divides the composite target into a DGTD region and an asymptotic region. The DGTD method is used for the multi-scale and multi-object region, which needs to be solved precisely, and the TDPO algorithm is utilized to calculate the electrically large rough sea surface quickly. As a result of the TDPO method, the shadow region and free space do not need to discretize in a large space; it greatly reduces the computational memory requirement and improves the computational speed. The numerical results show that the NRMSD of the DGTD method in the time domain, frequency domain and spatial domain can reach below 0.0685—that is, the accuracy of DGTDPO for multi-scale and multi-target regions is good enough. Compared with the DGTD method, the CPU time and memory requirement of DGTDPO can be reduced by 99.46% and 70.27% respectively, or even more. Meanwhile, the NRMSD of the time-domain, high-frequency approximation methods is over 0.2, and DGTDPO's NRMSD is only 0.0971. That is, the accuracy of the hybrid method is more than 64% higher than that of the approximate methods. Numerical results demonstrate that, compared with the DGTD method, the hybrid DGTDPO algorithm is more efficient in various composite scattering scenarios, and has better accuracy than time-domain, high-frequency approximation methods, including TDSBR and TDPO methods. Therefore, the hybrid method is of great significance for the scattering calculation of multi-scale and multi-target composite targets. Additionally, the DGTDPO method is also vital for the simulation of electromagnetic environments in engineering applications. Moreover, further work will also be devoted to more complex background environments, such as the attenuation of electromagnetic wave in various meteorological environments, and electromagnetic scattering in complex geographical environments, such as canyons, forests and so on.

**Author Contributions:** Conceptualization, J.Z., Y.W. and C.M.; methodology, J.Z. and C.M.; validation, J.Z. and C.M.; investigation, J.Z. and Y.W.; writing—original draft preparation, C.M.; writing—review and editing, J.Z. and Y.W. All authors have read and agreed to the published version of the manuscript.

**Funding:** This research was funded by the National Natural Science Foundation of China—the Joint Fund for Basic Research of High Speed Railway—Research on the Electromagnetic Environment and System Level EMC Theory and Applications to High Speed Railway (number U1734203); Basic Research Business Expenses of Beijing Jiaotong University—Special Project of Frontier Science Center of Smart High Speed Railway System (number 2020JBZD010); and Enterprise project (number W20L00370).

**Acknowledgments:** The authors wish to thank the reviewers for their valuable comments and suggestions concerning this manuscript.

**Conflicts of Interest:** The authors declare no conflict of interest.

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
