# Peer review of "A Fast, Hybrid, Time-Domain Discontinuous Galerkin-Physical Optics Method for Composite Electromagnetic Scattering Analysis"

_applsci, doi:10.3390/app11062694_

Round 1

Reviewer 1 Report

There are some weaknesses through the manuscript which need improvement. Therefore, the submitted manuscript cannot be accepted for publication in this form, but it has a chance of acceptance after a major revision. My comments and suggestions are as follows:

1- Abstract gives information on the main feature of the performed study, but some details about the utilized methods must be added.

2- Authors must clarify necessity of the performed research. Objectives of the study, and also differences with the previous researches must be clearly mentioned in the last part of introduction.

3- The literature study must be enriched. In this respect, authors must read and refer to the following papers: (a) https://doi.org/10.1002/nme.4936 (b) https://doi.org/10.1007/s40094-016-0217-9 (c) https://doi.org/10.1002/aenm.201900516

4- The main reference of each formula must be cited. Moreover, each parameters in equations must be introduced. Please double check this issue.

5- Standard deviation is the presented curves must be discussed.

6- In its language layer, the manuscript should be considered for English language editing. There are sentences which have to be rewritten.

7- The conclusion must be more than just a summary of the manuscript. List of references must be updated based on the proposed papers. Please provide all changes by red color in the revised version.

Reviewer 2 Report

Paper can be published

Reviewer 3 Report

This paper proposes a hybrid discontinuous Galerkin time domain (DGTD) and time domain physical optics (TDPO) numerical algorithm to accelerate solving transient scattered field in a multi-scale environment, where a local complex shaped scatter and a flat PEC scatter far from the local scatter are involved. The scattered field from the local complex shaped scatter is computed with DGTD, and its far field is acquired by using a Huygens surface. The scattered field from the far flat PEC scatter along with the interaction between the DGTD region and the flat PEC scatter is computed using TDPO, where a semi-analytical expression can be utilized to save computer resource.

From an engineering orientated perspective, the proposed algorithm is useful in the sense that it provides an approximate solution with acceptable accuracy. The authors claim that the PEC scatter in the asymptotic TDPO region is far from the DGTD region. It is unclear for the readers to determine how far is the distance between these two regions is enough to guarantee the accuracy of the approximate solution from the hybrid algorithm. Analysis on the dependency of the accuracy to the distance between the two computational regions should be supplemented.

There are a few typo or unclear statements in this manuscript.

1). In line 135:  …, and nmk is defined as the unit normal vector to the surface…

The symbol for this normal vector should have an arrow above, to be consistent with the symbol in Fig. 2. In addition, in Fig. 2, the normal vector should be nmk rather than nkm.

2). In Eq. (2), the second term on the right-hand side of the equation should have vhm rather than vem.

3). In Eqs. (2)-(5), the meaning of each term should be explained more lucidly. For example, electric and magnetic conductivity se and sh; the basis function terms Nqm, etc...

4). In Eqs. (7)-(14), the t and t symbols are rather confusing. Take Eq. (7) as an example, the right-hand side of the equation does not have t as a variable. Why does the left-hand side have t dependence? Are t and t the same?

5). In line 197, the last sentence. I believe what the authors wanted to say is that (11) represents the integral in (10) numerically. The same correction should be made to line 208-209.

6). In Eq. (17), what does the index m represent?

7). In the titles of Table 3 and Table 4, the term versus is inappropriate. It should be the NRMSD of….

8). In Eq. (18), what does S(k) represent? How could the model of the rough sea surface be incorporated in the proposed algorithm to replace the PEC scatter?

Round 2

Reviewer 1 Report

The paper has been improved and corresponding modifications have been conducted.

In my opinion, the current version can be considered for publication.